# Association of Protein Intake with Sarcopenia and Related Indicators Among Korean Older Adults: A Systematic Review and Meta-Analysis

**DOI:** 10.3390/nu16244350

**Published:** 2024-12-17

**Authors:** Minjee Han, Kyungsook Woo, Kirang Kim

**Affiliations:** 1Department of Food Science and Nutrition, Dankook University, Cheonan 31116, Republic of Korea; minjee@schmc.ac.kr (M.H.); jeeye798@gmail.com (K.W.); 2Institute of Health and Society, Hanyang University, Seoul 04763, Republic of Korea

**Keywords:** protein, sarcopenia, aged, Korean, meta-analysis, systematic review

## Abstract

Objectives: Due to variations in the standards for optimal protein intake and conflicting results across studies for Korean older adults, this study aimed to quantitatively integrate existing research on the association of protein intake with sarcopenia and related indicators in Koreans aged 65 and older through meta-analysis. Methods: A total of 23 studies were selected according to the study selection criteria (PICOS). Sixteen cross-sectional studies, 5 randomized controlled trials (RCTs), and 2 non-RCTs were included in the review, with 9 out of 23 studies included in the meta-analysis. We used fixed-effects models and performed subgroup and sensitivity analyses. Results: A meta-analysis found that the risk of sarcopenia was significantly higher in the <0.8 g/kg/day protein intake group compared to the 0.8–1.2 g/kg/day and ≥1.2 g/kg/day groups, with odds ratios (ORs) of 1.25 (95% confidence interval (CI), 1.10 to 1.42; *I*^2^ = 55%) and 1.79 (95% CI, 1.53 to 2.10; *I*^2^ = 71%), respectively. For low hand grip strength (HGS), the risk was higher in the <0.8 g/kg/day group compared to the 0.8–1.2 g/kg/day or ≥1.2 g/kg/day groups (OR 1.31; 95% CI, 1.03 to 1.65; *I*^2^ = 28%). No significant associations were found with other sarcopenia indicators, such as skeletal muscle mass, short physical performance battery score, balance test, gait speed, and timed up-and-go test. Conclusions: Lower protein intake is associated with a higher risk of sarcopenia and low HGS in Korean older adults. To establish protein intake recommendations for the prevention and management of sarcopenia in this population, further well-designed RCTs incorporating both protein supplementation and resistance training are necessary.

## 1. Introduction

The increasing incidence of various diseases due to aging is evident [1], with musculoskeletal disorders being prominently featured among them. Muscle mass decreases by 1–2% annually after the age of 50, while muscle strength decreases by 1.5% per year between the ages of 50 and 60, followed by a 3% decrease thereafter [2]. The decrease in muscle mass and strength are key factors in sarcopenia. Sarcopenia, beyond changes in the musculoskeletal system due to aging, is a complex and multi-factorial disease [3], influenced by genetic factors [4], hormonal changes (serum testosterone) [5], insulin resistance [6], and more. The prevalence of sarcopenia varies depending on the diagnostic criteria used. However, a meta-analysis reported in 2022 indicated that the global prevalence among individuals aged 60 and above ranges from 10% to 27%, while the prevalence among Asians was reported to be 15% [7]. In the 2022 National Health and Nutrition Survey conducted by the Korea Disease Control and Prevention Agency, sarcopenia was found to affect 7.9% of older adults aged 65 and above, with the prevalence increasing with age [8]. Age-related sarcopenia is an important issue as it is closely associated with falls [9], frailty and disability [10], and higher mortality rates [11].

Nutritional status can be considered a major contributing factor, particularly as reduced protein intake is closely associated with skeletal muscle loss [12], leading to consistent research efforts both domestically and internationally. Although it is known that protein or amino acid supplementation is effective for muscle protein synthesis in older adults [13], there is variation in the standards for optimal protein intake across studies. A review study suggested that consuming protein over 0.8 g/kg/day is generally associated with reducing muscle mass loss [12], but a cross-sectional study on Danish individuals aged 65 and above reported that there were no differences in skeletal muscle mass between older adults eating more than the recommendations (0.83 g/kg/day) and those eating less protein [14]. A cross-sectional study on Swedish women aged 65–70 suggested a minimum protein intake of 1.1 g/kg/day for preventing muscle loss [15]. A protein intake intervention study targeting Koreans aged 70–85 revealed that consuming 1.5 g/kg/day was more effective in preventing sarcopenia compared to 0.8 g/kg/day or 1.2 g/kg/day [16]. Due to anabolic resistance associated with aging, older adults require higher protein intake than younger individuals, with a particular need for increased levels of essential amino acids (EAAs) to better preserve muscle mass [17]. Based on evidence from various international observational and intervention studies, as well as expert consensus, the Korean Society of Geriatric Medicine and the Korean Nutrition Society recently recommended a minimum protein intake of 1.2 g/kg/day for the prevention of sarcopenia in older adults [18].

For the assessment of sarcopenia, the International Working Group proposed appendicular fat-free mass and gait speed as diagnostic criteria for sarcopenia [3]. In 2019, the revised European Working Group on Sarcopenia in Older People (EWGSOP2) focused on low strength (handgrip strength (HGS), chair stand) and evaluated low muscle quantity (appendicular skeletal muscle mass (ASM), ASM/height^2^) and low performance (gait speed, short physical performance battery (SPPB), timed up and go test (TUG), 400 m walk test) to diagnose sarcopenia [19]. Similarly, the Asian Working Group for Sarcopenia (AWGS) in their updated 2019 consensus, recommended assessing muscle strength (HGS), physical performance (6 m walk, 5-time chair stand, SPPB), and ASM [20]. Despite the cut-off points differing, the multiple parameters that are generally similar across these guidelines were assessed together to assess sarcopenia. A recent large-scale Korean elderly cohort study showed that protein intake was associated with sarcopenia assessed by the AWGS criteria [21], but studies examining the relationship between protein intake and each indicator used to assess sarcopenia such as skeletal muscle mass [22], HGS [23], and physical performance [24] among Korean older adults did not find significant correlations.

Thus, because of a lack of evidence for the effect of protein intake on the individual parameter of sarcopenia, especially for Korean older adults, an updated systematic review and meta-analyses need to be conducted. Therefore, this study aimed to quantitatively integrate existing research on protein intake and its association with sarcopenia and related indicators among Koreans aged 65 and above through a meta-analysis in order to understand the current state and clarify the relationship between protein intake and sarcopenia.

## 2. Subjects and Methods

### 2.1. Search Strategy

This study was conducted following the PRISMA 2020 statement guidelines and the checklist is in the Appendix A [25]. To ensure systematic literature selection, we defined the key questions using the Population, Intervention, Comparison, Outcome, and Study design (PICOS) framework and used these as search terms for identifying relevant literature. The search terms used are as follows and were based on the PICOS criteria: (“aged” OR “elder*” OR “older*” OR “korea” OR “korea*”) and (“proteins” OR “proteinous” OR “protein intake*” OR “protein consumption*” OR “protein*” OR “amino acid*” OR “amino acids, branched chain” OR “branched chain amino acid*” OR “BCAA*” OR “leucine” OR “HMB” OR “β-hydroxy-β-methylbuty*” OR “beta hydroxyisovaleric acid”) and (“muscles” OR “molecular weight” OR “muscle*” OR “muscle mass” OR “lean body mass” OR “lean mass” OR “fat free mass” OR “muscle strength” OR “muscle strength*” OR “muscle force*” OR “muscle power*” OR “hand strength” OR “hand strength*” OR “hands strength*” OR “hand force*” OR “hand power*” OR “hand grip*” OR “hands grip*” OR “gait speed*” OR “gait velocit*” OR “gait analy*” OR “gait*” OR “walking speed” OR “walking speed*” OR “SARC-F questionnaire*” OR “SARC-F*” OR “chair stand test*” OR “chair stand*” OR “standing position*” OR “standing posture*” OR “short physical performance battery” OR “SPPB” OR “physical functional performance” OR “time up and go test*” OR “timed up and go test*” OR “timed up and go” OR “timed up & go” OR “TUG” OR “6 min walking test*” OR “6 min walk test” OR “6-min walk test*” OR “walk test” OR “walk test*” OR “walking test*” OR “six min walk test*” OR “six min walking test*” OR “SMWT*” OR “six-min walk*” OR “6MWT” OR “sarcopenia” OR “frail elderly” OR “frailty” OR “frail*” OR “frailty index” OR “frailty indic*” OR “frail index” OR “frail indic*”). MeSH terms and Emtree were used to enhance sensitivity and specificity, with searches focused on titles and abstracts. We used database filters to select studies published in English and those related to humans. The study design included all types of research analyzing the association between protein intake and sarcopenia. Two researchers (MH and KS) independently searched the literature. We conducted a literature search using core electronic databases (MEDLINE via PubMed, EMBASE, Cochrane) and Korean databases (KoreaMed, RISS, ScienceON) to identify publications up to September 2022. Additionally, we reviewed the references of the identified literature for additional relevant studies.

### 2.2. Inclusion and Exclusion Criteria

The inclusion criteria were as follows: (1) studies targeting Korean adults aged 65 and older (Population), (2) studies focusing on protein intake as the exposure (Intervention), (3) studies comparing protein intake levels between groups (Comparison), (4) studies reporting outcomes related to sarcopenia or sarcopenia-related indicators (Outcome), and (5) studies that measured the risk of sarcopenia and enabled quantitative synthesis. The exclusion criteria were as follows: (1) studies not targeting Koreans or including age groups younger than 65 years, or studies targeting animals or microbes instead of humans (Population); (2) studies which the intervention in the experimental group was not protein, or studies comparing the effects of interventions other than protein (Intervention); (3) studies that could not classify protein intake levels into three groups (<0.8 g/kg/day, 0.8–1.2 g/kg/day, ≥1.2 g/kg/day) and therefore could not be compared between groups (Comparison); (4) studies that presented outcomes other than sarcopenia or sarcopenia-related indicators (Outcome); (5) case reports, articles published solely as abstracts, qualitative studies, literature reviews, and meta-analyses; and (6) publications in languages other than Korean or English.

### 2.3. Risk of Bias Assessment

Two researchers (MH and KS) independently assessed the risk of bias using the QualSyst (Checklist for Assessing the Quantitative Studies) tool, and in the case of discrepancies in the results, consensus was reached through discussion with a co-researcher (KK). QualSyst, developed by the Alberta Heritage Foundation for Medical Research (AHFMR) in Canada, is a validated tool known for its usefulness and validity, and it can be applied to various research designs. Therefore, it is highly suitable for use, even in studies that include literature from multiple research designs, such as the present study [26,27]. The evaluation consisted of 14 items assessing the “sufficient description of research questions and objectives, appropriate study design, appropriateness of participant selection method, description of participant characteristics, randomization, blinding of researchers to interventions, blinding of participants to interventions, clear measurement of outcomes and exposures, appropriateness of sample size, appropriateness of analysis methods, reporting of estimates of outcome variability, control of confounding variables, sufficient description of results, and conclusions supported by the results”. These criteria were adapted as literature selection criteria using a cut-off score of 0.55, as provided by the AHFMR [28].

### 2.4. Data Extraction

The literature were preliminarily reviewed to select extraction items, and two researchers (MH and KS) independently extracted data from each study and conducted cross-checks to enhance accuracy. The extraction information includes subjects (number of subjects, gender, age), group (intervention, placebo group), protein intake (protein intake level, duration), and outcomes (sarcopenia or sarcopenia-related indicators including sarcopenic obesity (SO), muscle mass, HGS, SPPB score, balance test, gait speed, and TUG test). In some cases, a conversion process was undertaken to standardize criteria and units for quantitative synthesis. The process was reviewed and compared, and where discrepancies were noted, consensus was reached via discussion or consultation with a co-researcher (KK).

The classification of protein intake levels was based on the estimated average requirement (EAR) of 0.8 g/kg/day as suggested by the 2020 Korean Dietary Reference Intakes (KDRIs) [29] and the recommendation of 1.2 g/kg/day by the Korean Society of Geriatric Medicine and the Korean Nutrition Society [18]. The studies were then categorized into three groups (<0.8 g/kg/day, 0.8–1.2 g/kg/day, and ≥1.2 g/kg/day). In cases when only the total protein intake was provided in the study, it was converted to grams of protein per unit body weight by dividing by the body weight. If the body weight of the participants was not reported, the average body weight for the relevant age group from the Korean National Health and Nutrition Examination Survey (KNHANES) data was applied. The calculated values for outcome variables included the number of subjects with events (N) and percentage (%), odds ratio (OR), 95% confidence interval (CI), mean, standard deviation (SD), or standard error (SE) as extracted from each study.

### 2.5. Statistical Analysis

The results extracted from each study were synthesized for each outcome variable, which was classified into the number of subjects with events (unadjusted OR), continuous data (mean, SD), and multivariable adjusted data (adjusted OR). The effect sizes for outcomes (indicators related to sarcopenia) were reported to vary, so the pooled effects were estimated separately for each type of sarcopenia indicators. Sarcopenia criteria varied across the literature but were defined according to the researchers’ definitions. HGS was categorized by classifying subjects with low HGS according to a specific threshold for analysis. If a single literature presented results separately for men and women, each was considered as an individual study for analysis. Due to the limited number of synthesized studies from one country in this study, which imply small values between study variance, a fixed-effects model was used for the analysis model [30,31].

We estimated ORs and 95% CI as the pooled effects for dichotomous data using the Mantel–Haenszel method [32]. The effect synthesis for continuous data used weighted mean differences [33]. The multivariable adjusted data (OR, 95% CIs) were synthesized using the generic inverse variance method [33]. The ORs and 95% CIs were log-transformed to calculate pooled ORs with 95% CIs, which were then converted back to ORs for interpretation [34]. The pooled effect estimate was calculated using the inverse variance estimation method, where the reciprocal of the variance for the effect size obtained from individual studies was applied as the weight [33]. For conducting a more detailed analysis of the data and evaluating the effects within specific subgroups, subgroup analyses were performed by dividing the studies into subgroups based on general characteristics. To enhance the overall reliability of the study results, sensitivity analyses were conducted by excluding studies deemed most likely to have the greatest impact on the analysis, and changes in effect size were observed [35,36]. To visualize and summarize the association between exposure and outcome, we presented the information as forest plots. Publication bias was visually assessed using funnel plots [37]. This meta-analysis was conducted using R software (version 4.2.2).

### 2.6. Ethic Statement

This study used data from previously published papers and meta-analysis. Therefore, institutional review board approval was not deemed necessary.

## 3. Results

### 3.1. Studies Included in Analysis

The study selection process is detailed in Figure 1. A total of 2691 articles were retrieved from 6 databases. After removing duplicates, 1574 articles were screened based on their titles and abstracts according to the PICOS criteria. Among them, 1542 irrelevant articles were excluded. Thirty-two articles were assessed for eligibility through full-text review. Four articles were excluded because only the abstracts were available and nine articles were excluded because the information for subjects aged 65 and older could not be determined. After citation searching, 4 additional articles were included, resulting in 23 articles selected for the systematic review. Of these, 14 articles were excluded due to insufficient data or the inability to synthesize the data, and finally, 9 articles were included in the meta-analysis.

### 3.2. Study Characteristics

#### 3.2.1. Association Between Protein Intake and Sarcopenia

The characteristics of the studies included in the systematic review and meta-analysis on protein intake with sarcopenia are described in Table 1. Eight cross-sectional studies were included, all of which were published after the year 2000. Six of these studies used data from the KNHANES, of which 5 studies [38,39,40,41,42] utilized data from 2008 to 2011. Protein intake was evaluated using either a 24 h recall method or food frequency questionnaire (FFQ). The definition of sarcopenia varied across studies. Two studies [21,43] defined sarcopenia by assessing both muscle mass and muscle strength according to the criteria recommended by AWGS 2014. One study [41] based its definition on AWGS 2014 criteria, but used only muscle mass without assessing muscle strength. Five studies evaluated only muscle mass. Among these, two studies [39,44] defined sarcopenia as muscle mass being 2 SD below the mean, adjusted for body weight, compared to a healthy young adult reference group. Although the Huh Y [39] study did not explicitly define sarcopenia but referred to it as low muscle mass, it effectively used the same criteria for evaluating sarcopenia as the other studies. Therefore, in this study, it is classified as sarcopenia. In contrast, three studies [38,40,42] used the definition of 1 SD below the mean. Dual-energy X-ray (DXA) was used to measure muscle mass as ASM in all studies, and muscle strength was assessed using HGS.

Out of a total of eight studies, three studies [21,42,43] were excluded due to insufficient data, and the five remaining studies [38,39,40,41,44] were used in the meta-analysis. The total number of subjects included in the meta-analysis was 13,285.

#### 3.2.2. Association Between Protein Intake and Indicators Related to Sarcopenia

The characteristics of the studies included in the systematic review and meta-analysis on protein intake with sarcopenia-related indicators are described in Table 2. Expect for one publication, all were published after 2000. A total of 15 studies were analyzed, categorized by outcome indicators (SO, muscle mass, HGS, SPPB, balance test, gait speed, and TUG test). The analysis included eight cross-sectional studies using data from KNHANES from 2008 to 2019, along with five randomized controlled trials (RCTs) and two non-RCTs. The invention periods for the RCTs and non-RCTs ranged from 4 to 12 weeks. Among the RCTs and non-RCTs, three studies provided oral nutritional supplements (ONS) [23,45,46], two studies provided protein supplements [16,24], and two studies combined exercise with the provision of protein supplements and protein-rich foods (milk, egg), respectively [47,48]. Protein intake was evaluated using dietary records, the 24 h recall method, or an FFQ.

After excluding studies that could not be quantitatively synthesized, two or three studies per outcome indicator were included in the meta-analysis. The three studies [42,49,50] included in the SO analysis assessed ASM and body mass index (BMI). Muscle mass was measured differently across studies, including skeletal muscle mass (kg) [24,47], lean body mass (LBM) (kg) [45,46], muscle mass (kg) [48,51], ASM (kg) [16], and ASM/Wt (%) [51]. HGS was also measured in various ways, such as relative grip strength (HGS/Wt) [24,53], HGS (kg) [16,23,43,45,52,54], and risk for low HGS (OR) [55,56]. In this study, a meta-analysis was conducted using low HGS. Two studies analyzing low HGS were included. One study [55] was based on the 2019 AWGS guidelines, while the other applied the cut-off values proposed in a study conducted on Koreans using KNHANES data from 2014 to 2015 [57]. Indicators of physical performance, such as SPPB, balance tests, gait speed, and TUG test, were used as measures related to sarcopenia in the studies.

### 3.3. Risk of Bias Assessment and Results of Systematic Review

Using the QualSyst tool, 14 criteria were used, assigning 2 points for full compliance (Yes), 1 point for partial compliance (Partial), and 0 points for non-compliance (No), to calculate a comprehensive score. The average score of the 23 studies was 0.90 points (range: 0.58 to 1.00), indicating that all included studies exceeded the cut-off point of 0.55 points. For the individual criteria, in the research design category, all 16 cross-sectional studies received a “Partial” rating. The study on the fracture patients [43] and two non-RCT studies [46,48] received a “Partial” rating for the participant selection method. Three studies [43,46,48] did not provide sufficient details about the participant characteristics. Four studies [24,46,47,48] did not provide adequate data to assess the appropriateness of the sample size. Two studies [46,48] had insufficient control of confounding variables, while six studies [38,41,43,44,47,53] did not consider confounding variables at all. Nevertheless, the overall quality of the studies evaluated was considered high.

Among the eight cross-sectional studies analyzing the association between protein intake and sarcopenia, five studies [21,38,39,40,41] found that the risk of sarcopenia was significantly higher in the low protein intake group compared to the high protein intake group. In the three studies, although not statistically significant, lower protein intake was observed among individuals with sarcopenia (Table 1). Indicators related to sarcopenia included SO, muscle mass, HGS, SPPB score, balance test, gait speed, and TUG test (Table 2). The review included eight cross-sectional studies, five RCTs, and two non-RCTs. For the outcome measure, all three cross-sectional studies included in the analysis of SO showed a statistically significant association with lower protein intake [42,49,50]. In the muscle mass studies, all four RCTs [16,24,45,47] reported significant increases in muscle mass in the protein intervention groups. In two non-RCT studies, significant results were found only in studies [48] that combined exercise with protein intake. Additionally, one cross-sectional study [51] reported a positive correlation between protein intake and muscle mass. Regarding HGS, four RCTs and six cross-sectional studies examined the association. In two RCTs, significant improvements in relative grip strength and HGS were observed in the intervention groups [16,24]. In the cross-sectional studies, all but one of the six studies reported an association between protein intake and HGS [43,52,53,54,56]. Studies analyzing the association with physical performance included three RCTs for each indicator. Significant results were found in two RCTs for the SPPB score, gait speed, and TUG test [16,23], while the balance test showed improvement in the intervention group in only one study [16].

### 3.4. Meta-Analysis Results

#### 3.4.1. Meta-Analysis of the Association Between Protein Intake and Sarcopenia

The results of a meta-analysis on the association between protein intake levels and sarcopenia showed significant difference in the risk of sarcopenia between the <0.8 g/kg/day protein intake group and the 0.8–1.2 g/kg/day protein intake group (OR = 1.25; 95% CI, 1.10 to 1.42; *I*^2^ = 55%). The risk of sarcopenia in the <0.8 g/kg/day protein intake group compared to the ≥1.2 g/kg/day protein intake group was estimated to be significantly higher, with an odds ratio of 1.79 (95% CI, 1.53 to 2.10; *I*^2^ = 71%). The heterogeneity was high at 71%, so we reanalyzed the data using a random-effects model. The results showed similar results, showing significant differences (OR = 1.67; 95% CI, 1.18 to 2.37; *I*^2^ = 71%). It was also determined that the risk of sarcopenia was higher in the 0.8–1.2 g/kg/day protein intake group compared to the ≥1.2 g/kg/day protein intake group (OR = 1.30; 95% CI, 1.11 to 1.52; *I*^2^ = 65%). The results analyzed with adjusted OR were similar. The risk of sarcopenia in the <0.8 g/kg/day protein intake group compared to the ≥1.2 g/kg/day protein intake group was OR 1.87 (95% CI, 1.46 to 2.38; *I*^2^ = 42%), and a significant risk was also observed in the 0.8–1.2 g/kg/day protein intake group with an OR of 1.58 (95% CI, 1.25 to 2.01; *I*^2^ = 31%) (Figure 2).

The results of the subgroup analysis by gender are shown in Figure 3. There was no significant difference in the risk of sarcopenia among women with unadjusted OR. For the adjusted OR values, the risk between the 0.8–1.2 g/kg/day protein intake group and the ≥1.2 g/kg/day protein intake group was significantly higher at OR 2.01 when analyzed for men only (95% CI, 1.36 to 2.99; *I*^2^ = 53%).

#### 3.4.2. Meta-Analysis of the Association Between Protein Intake and Indicators Related to Sarcopenia

The analysis of protein intake levels and indicators related to sarcopenia synthesized two or three effect estimates for each outcome variable. Statistically significant differences were observed only in low HGS. The risk of low HGS was higher in the <0.8 g/kg/day protein intake group compared to the 0.8–1.2 g/kg/day or ≥1.2 g/kg/day protein intake groups, with an OR 1.31 (95% CI, 1.03 to 1.65; *I*^2^ = 28%). The results for the skeletal muscle mass, SPPB score, balance test, gait speed, and TUG test, analyzed by calculating the effect sizes using the mean differences, were not significant (Figure 4).

### 3.5. Publication Bias and Sensitivity Analyses

To verify publication bias among the literature, data were categorized by type, and funnel plots were used to visually assess the degree of asymmetry (Figure 5). Although showing an asymmetric tendency, statistical verification through an Egger’s regression test indicated *p*-values of 0.16, 0.48, and 0.09, respectively, suggesting no evidence of publication bias. In the sensitivity analysis of high protein intake and sarcopenia, which excluded studies with the greatest impact on the results, revealed that the risk of sarcopenia increased with unadjusted OR (<0.8 vs. 0.8–1.2), adjusted OR (<0.8 vs. ≥1.2), and adjusted OR (0.8–1.2 vs. ≥1.2), with an OR of 1.32 (95% CI, 1.15 to 1.51; *I*^2^ = 0%), 2.51 (95% CI, 1.75 to 3.59; *I*^2^ = 0%), and 1.93 (95% CI, 1.33 to 2.80; *I*^2^ = 21%), respectively (Figure 6).

## 4. Discussion

This study conducted a meta-analysis by integrating primary studies targeting Koreans aged 65 and older to examine the association between protein intake levels and sarcopenia or sarcopenia-related indicators. The risk of sarcopenia was significantly higher in the <0.8 g/kg/day protein intake group compared to the group with 0.8–1.2 g/kg/day or ≥1.2 g/kg/day protein intake, and the risk of low HGS among the indicators related to sarcopenia was also significantly higher in the group with <0.8 g/kg/day protein intake.

The results of our study were similar to those reported in a meta-analysis targeting multinational older populations in Europe, India, Australia, and other countries, which indicated that lower protein intake was associated with sarcopenia [58]. The Health ABC (Health, Aging, and Body Composition) study [59], which investigated the association between protein intake and LBM over a 3-year period, also reported findings consistent with our study. They found that older adult individuals in the highest quintile of protein intake (1.2 g/kg/day) experienced 40% less skeletal muscle loss compared to those in the lowest quintile (0.8 g/kg/day) [59].

The indicators related to sarcopenia showed varying results depending on the specific measures. The risk of low HGS was associated with low protein intake. However, this analysis was based on only two cross-sectional studies. Due to the nature of cross-sectional studies, there are inherent limitations in inferring causality. On the other hand, the means of skeletal muscle mass, SPPB score, balance test, gait speed, and TUG test, analyzed through two RCTs as other indicators related to sarcopenia, showed no significant association with protein intake. RCTs are considered the gold standard for intervention studies and can better establish causality, but longer intervention periods are required to observe meaningful outcomes. According to previous studies, the SPPB score increased significantly with longer intervention periods (≥24 weeks) [60]. However, the RCTs included in our study had relatively short intervention periods ranging from 4 to 12 weeks, which might explain the lack of significant associations observed. Additionally, the absence of resistance exercise in the interventions of the included RCTs may have further limited the effects of protein supplementation. When protein intake is combined with repetitive resistance exercise, net protein balance increases and muscle protein accretion is promoted over time [61]. Several studies have reported a positive association between protein intake and changes in HGS [62,63]. However, a meta-analysis of 10 RCTs reported that protein supplementation alone, without resistance exercise, may not induce changes in HGS [64]. A recent meta-analysis published in 2024 also found that HGS (kg) and appendicular skeletal muscle mass index (ASMI, kg/m^2^) improved when whey protein supplementation was combined with resistance training [65]. Similarly, a meta-analysis examining the effects of protein supplementation combined with resistance exercise on physical function in older adults reported significant improvements in SPPB scores [66]. These findings underscore the need for long-term studies targeting Korean older adults, which incorporate both protein supplementation and resistance exercise. Such studies are essential to comprehensively evaluate the combined effects of these interventions on sarcopenia-related outcomes and provide more conclusive evidence for effective management strategies.

Skeletal muscle serves as a major reservoir of amino acids [13], and protein intake enhances muscle protein synthesis. Therefore, sufficient protein intake is necessary for older adults at high risk of sarcopenia [60]. Moreover, due to the decreased protein anabolic response and the presence of catabolic conditions associated with acute and chronic diseases, older adults require higher protein intake compared to younger adults [67]. However, many older adults often have comorbidities such as kidney disease, liver disease, or cancer, which complicates protein intake recommendations. For instance, the 2024 Kidney Disease: Improving Global Outcomes guidelines recommend maintaining protein intake at 0.8 g/kg/day for individuals with chronic kidney disease and avoiding high protein intake (>1.3 g/kg/day) for those at risk of chronic kidney disease progression [68]. Similarly, in chronic liver diseases such as cirrhosis, protein intake is restricted to prevent hepatic encephalopathy [69]. Balancing these recommendations is challenging, particularly as sarcopenia is a major complication that significantly worsens the prognosis of liver cirrhosis patients. Traditionally, guidelines recommend a protein intake of 0.8 g/kg/day for adults regardless of age or gender [67]. Similarly, the 2020 KDRIs also suggest an EAR of 0.8 g/kg/day and a recommended nutrient intake (RNI) of 1.0 g/kg/day for individuals aged 65 and older [29]. However, these guidelines do not address the specific needs of aging populations or the risk of sarcopenia. Therefore, it is crucial to develop evidence-based, appropriate protein intake guidelines for Korean older adults. These guidelines should recommend a tailored approach for geriatric patients with acute or chronic diseases, tailored to the specific needs of each individual.

Recently, the PROT-AGE Study Group of the European Union Geriatric Medicine Society (EUGMS) recommended an optimal protein intake for older adults of 1.0–1.2 g/kg/day, higher than that for younger adults [67]. Additionally, the Sarcopenia and Physical Frailty in Older People: Multi-component Treatment Strategies (SPRINTT) RCT, which followed 1500 older adults aged 70 and above with physical frailty and sarcopenia for 3 years, also recommends a protein intake of at least 1.0–1.2 g/kg/day to prevent sarcopenia [70]. In this meta-analysis, the group with a protein intake of less than 0.8 g/kg/day had a higher risk of sarcopenia compared to the group with a protein intake of 0.8 g/kg/day or greater. Therefore, it can be inferred that a protein intake of 0.8 g/kg/day or greater may be beneficial for preventing and managing sarcopenia in Korean older adults. Additionally, the analysis of the risk of sarcopenia between the protein intake group of 0.8–1.2 g/kg/day and the group with protein intake over 1.2 g/kg/day showed that the 0.8–1.2 g/kg/day group had a 1.3 times higher risk of sarcopenia compared to the group with over 1.2 g/kg/day. However, due to the limited number of studies included in the meta-analysis, additional research is needed to establish evidence-based specific guidelines for protein intake levels tailored to Korean older adults.

Previous studies have reported that the food source of protein intake, along with the level of protein consumption, would be important to sarcopenia management. The Health ABC study reported an association between animal protein intake and preservation of LBM [59], and a meta-analysis published in 2021 found that animal protein intake tends to be more beneficial for preserving LBM compared to plant protein intake [71]. However, a diet rich in animal protein sources can also lead to increased energy and fat intake, which may potentially impact obesity and other chronic diseases [72]. A characteristic feature of sarcopenia in Korean older adults is not only insufficient protein intake but also their reliance on plant-based foods, such as soybeans and tofu, rather than animal-based foods. Therefore, understanding the impact of protein sources on Korean older adults is important. While animal-based protein may be more effective in preventing and managing sarcopenia, excessive intake could increase the risk of various chronic diseases such as obesity, colorectal disease, and cardiovascular diseases. On the other hand, plant-based protein may help prevent chronic diseases. Thus, to establish appropriate protein intake recommendations for Korean older adults, it is essential to determine the optimal ratio of animal-based to plant-based protein. Therefore, further research on this topic is needed.

This study has several limitations. First, the diagnostic criteria for sarcopenia differ across studies, so several criteria are mixed. In 2019, the EWGOSP2 developed sarcopenia diagnostic criteria using declines in muscle mass, strength, and physical function as indicators, but consensus has not yet been reached, and there is a lack of clear criteria for assessing muscle mass. In addition, this study included several studies conducted before 2019, and sarcopenia was classified according to the researcher’s definition criteria during this analysis. As a result, the differences in diagnostic criteria pose a risk of underestimating or overestimating the research outcomes. Even when synthesizing results, conclusions may be limited due to variations in diagnostic standards. Therefore, additional studies using new, consistent diagnostic criteria are necessary. Second, in some studies where information on the subject’s body weight was unavailable, weight values were adopted from the KNHANES. Due to the lack of standardized units for protein intake across studies, bias may arise during the standardization process, which could hinder the accurate reflection of differences between studies. Third, due to the two limitations discussed above, heterogeneity may have increased, which can be assessed for bias in the results through subgroup analysis [73]. Although there were not many variables for subgroup analysis, it was performed by sex, and the small number of studies made it difficult to interpret the causes or report reliable results. Fourth, some of the cross-sectional studies from the KNHANES of the same year were used for the meta-analysis. Although these studies had different analysis groups and analysis units of protein intake variables, the generalization of the results due to data duplication would be limited. Additionally, due to the nature of cross-sectional studies, there are limitations in establishing causal relationships.

## 5. Conclusions

In conclusion, this study showed that low protein intake levels of less than 0.8 g/kg/day may increase the risk of sarcopenia and low HGS in Korean older adults. In the future, based on the preceding limitations and considerations, well-designed RCT studies are needed to accumulate evidence to establish protein intake standards for the prevention and treatment of sarcopenia for Korean older adults.

## Figures and Tables

**Figure 1 nutrients-16-04350-f001:**
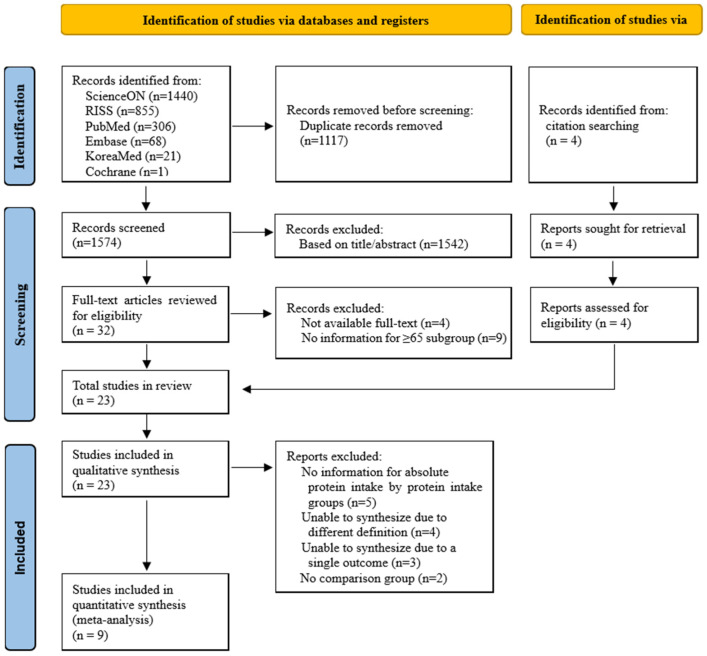
PRISMA 2020 flow diagram of article selection.

**Figure 2 nutrients-16-04350-f002:**
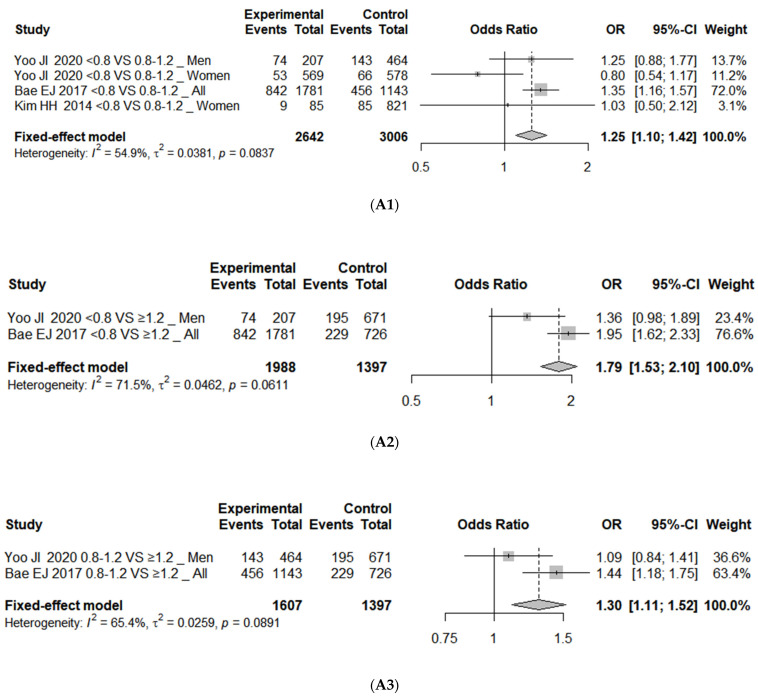
Forest plot of association between protein intake and (**A1**) sarcopenia with unadjusted OR (<0.8 vs. 0.8–1.2 as reference) [38,41,44], (**A2**) sarcopenia with unadjusted OR (<0.8 vs. ≥1.2 as reference) [38,41], (**A3**) sarcopenia with unadjusted OR (0.8–1.2 vs. ≥1.2 as reference) [38,41], (**B1**) sarcopenia with adjusted OR (<0.8 vs. ≥1.2 as reference) [38,39,40], and (**B2**) sarcopenia with adjusted OR (0.8–1.2 vs. ≥1.2 as reference) [38,39,40]. OR represents risk of each outcome in comparison group compared to reference group 0.8, protein intake 0.8 g/kg/day; 0.8–1.2, protein intake 0.8–1.2 g/kg/day; ≥1.2, protein intake ≥ 1.2 g/kg/day. OR, odds ratio; CI, confidence interval; SE, standard error.

**Figure 3 nutrients-16-04350-f003:**
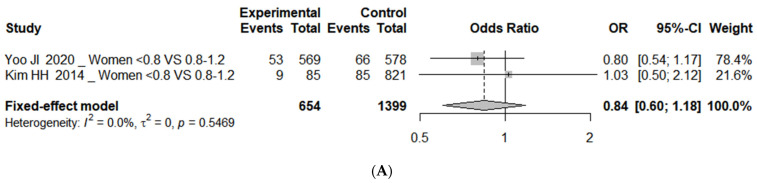
Forest plot of subgroup analysis of association between protein intake and (**A**) sarcopenia with unadjusted OR (<0.8 vs. 0.8–1.2 as reference) [41,44], and (**B**) sarcopenia with adjusted OR (0.8–1.2 vs. ≥1.2 as reference) [39,40]. OR represents risk of each outcome in comparison group compared to reference group 0.8, protein intake 0.8 g/kg/day; 0.8–1.2, protein intake 0.8–1.2 g/kg/day; ≥1.2, protein intake ≥ 1.2 g/kg/day. OR, odds ratio; CI, confidence interval; SE, standard error.

**Figure 4 nutrients-16-04350-f004:**
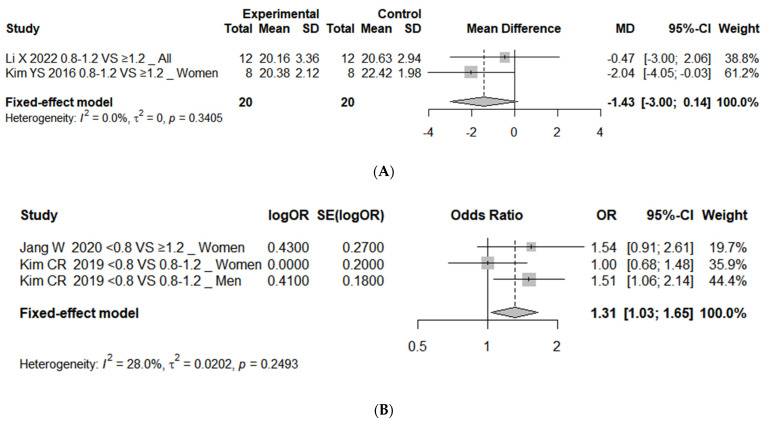
Forest plot of association between protein intake and (**A**) skeletal muscle mass (kg) (0.8–1.2 vs. ≥1.2 as reference) [24,47], (**B**) low HGS (<0.8 vs. 0.8–1.2 or ≥1.2 as reference) [55,56], (**C1**) SPPB score (<0.8 vs. 0.8–1.2 as reference) [16,23], (**C2**) SPPB score (0.8–1.2 vs. ≥1.2 as reference), (**D1**) balance test (seconds) (<0.8 vs. 0.8–1.2 as reference) [16,23], (**D2**) balance test (seconds) (0.8–1.2 vs. ≥1.2 as reference) [16,24], (**E**) gait speed (m/s) (<0.8 vs. 0.8–1.2 as reference) [16,23], and (**F**) TUG test (seconds) (<0.8 vs. 0.8–1.2 as reference) [16,23]. OR represents risk of each outcome in comparison group compared to reference group 0.8, protein intake 0.8 g/kg/day; 0.8–1.2, protein intake 0.8–1.2 g/kg/day; ≥1.2, protein intake ≥ 1.2 g/kg/day. HGS; hand grip strength; SPPB, short physical performance battery; TUG, timed up-and-go; SD, standard deviation; MD, mean difference; OR, odds ratio; SE, standard error; CI, confidence interval.

**Figure 5 nutrients-16-04350-f005:**
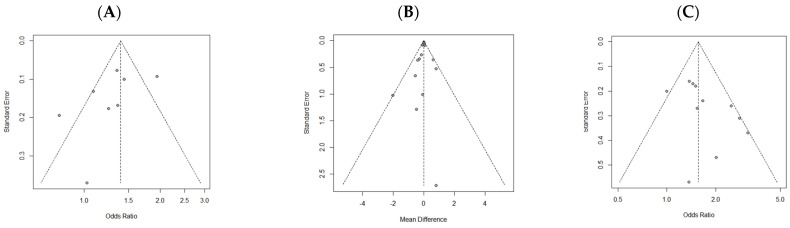
Funnel plot of included studies of (**A**) unadjusted data (sarcopenia), (**B**) continuous data (skeletal muscle mass, SPPB score, balance test, gait speed, and TUG test), and (**C**) adjusted data (sarcopenia, low HGS). SPPB, short physical performance battery test; TUG, timed up-and-go; HGS; hand grip strength.

**Figure 6 nutrients-16-04350-f006:**
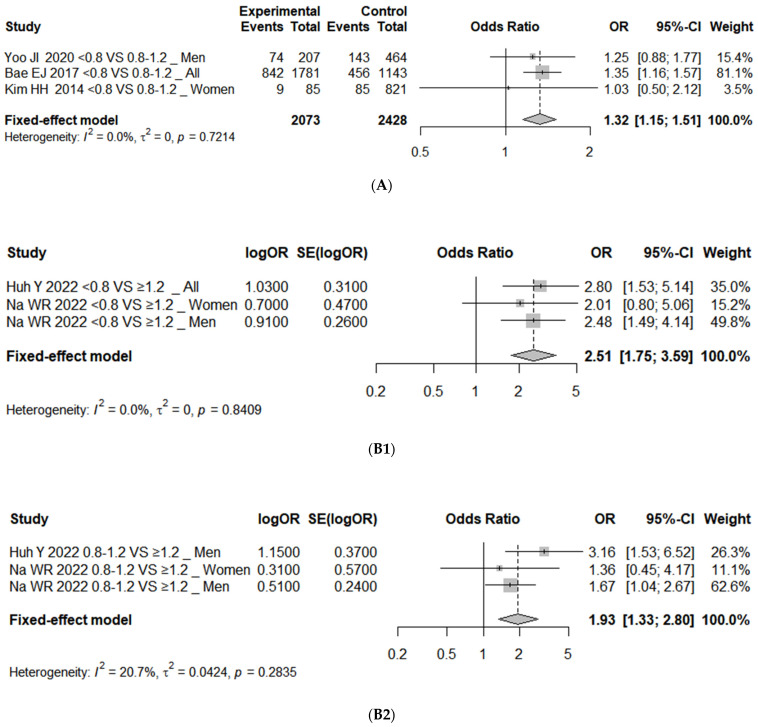
Forest plot of sensitivity analysis of association between protein intake and (**A**) sarcopenia with unadjusted OR (<0.8 vs. 0.8–1.2 as reference) [38,41,44], (**B1**) sarcopenia with adjusted OR (<0.8 vs. ≥1.2 as reference) [39,40], and (**B2**) sarcopenia with adjusted OR (0.8–1.2 vs. ≥1.2 as reference) [39,40]. OR represents risk of each outcome in comparison group compared to reference group 0.8, protein intake 0.8 g/kg/day; 0.8–1.2, protein intake 0.8–1.2 g/kg/day; ≥1.2, protein intake ≥ 1.2 g/kg/day. OR, odds ratio; CI, confidence interval; SE, standard error.

**Table 1 nutrients-16-04350-t001:** Characteristics of studies included in systematic review and meta-analysis on protein intake with sarcopenia.

Author (Year)	Study Design	Data Source	NO. of Subjects (Women/Men) Age (Years)	Exposure Variable Dietary Assessment Method	Outcome and Definition	Results	Main Finding	Meta-Analysis
Choi KA et al. (2022) [43]	Cross-sectional	Hip fracture patients	35/12, ≥65	Protein intake (g/d) FFQ	Sarcopenia: ASM (by DXA)/Ht(m)^2^) < 5.4 in women, <7.0 in men and HGS < 18 kg in women, <26 kg in men (according to AWGS 2014 criteria)	Lower protein intake marginally associated with sarcopenia (*p* = 0.19)	NS	Exclusion: No comparison group
Huh Y et al. (2022) [39]	Cross-sectional	KNHANES 2008–2011	358 (gender unknown) ≥65	Quartile of protein intake 24 h recall	Sarcopenia: ASM (by DXA)/Wt (%) < 2 SD below the mean of young health group without any chronic disease (20–39 years)	Q1 protein group’s OR 2.81, 95% CI 1.53–5.16 (ref. Q4)	Sarcopenia +	Inclusion
Na WR et al. (2022) [40]	Cross-sectional	KNHANES 2008–2011	1822/1414, ≥65	Protein intake (g/kg/d) 24 h recall	Sarcopenia: ASM (by DXA)/Wt (%) < 1 SD below the mean of young health group without any chronic disease (19–39 years)	Protein intake < 0.8 g/kg/d group’s OR 2.18, 95% CI 1.42–3.33 in women, OR 2.46, 95% CI 1.48–4.09 in men, OR 2.36, 95% CI 1.68–3.30 in all (ref. ≥ 1.2 g/kg/d)	Sarcopenia +	Inclusion
Park SJ et al. (2022) [21]	Cross-sectional	KFACS	416/385, 70–84	Protein intake (g/d) 24 h recall	Sarcopenia: ASM (by DXA)/Ht(m)^2^ < 5.4 in women, <7.0 in men, and HGS < 18 kg in women, <26 kg in men (according to AWGS 2014 criteria)	Q4 protein intake group’s OR 0.44, 95% CI 0.20–0.93 (ref. Q1)	Sarcopenia +	Exclusion: No data on protein intake groups
Yoo JI et al. (2020) [41]	Cross-sectional	KNHANES 2008–2011	2322/1698, ≥65	Protein intake (g/d) 24 h recall	Sarcopenia: ASM (by DXA)/Ht(m)^2^) < 5.4 in women, <7.0 in men	Lower protein intake in sarcopenia group compared to normal group in women and men (*p* < 0.001)	Sarcopenia +	Inclusion
Lim HS et al. (2018) [42]	Cross-sectional	NHANES 2008–2011	1642/1850, ≥65	Protein intake (g/d) 24 h recall	Sarcopenia: ASM (by DXA)/Wt (%) < 1 SD below the mean of young health group without any chronic disease (20–39 years)	Sarcopenia group tended to have lower protein intake compared to non-sarcopenia group.	NS	Exclusion: No data on protein intake groups
Bae EJ et al. (2017) [38]	Cross-sectional	KNHANES 2008–2011	2259/1642, ≥65	Protein intake (g/kg/d) 24 h recall	Sarcopenia: ASM (by DXA)/Wt (%) < 1 SD below the mean of young health group without any chronic disease (20–39 years)	Protein intake < 0.8 g/kg/d group’s OR 1.45, 95% CI 1.03–2.03 (ref. ≥ 1.2 g/kg/d)	Sarcopenia +	Inclusion
Kim HH et al. (2014) [44]	Cross-sectional	KNHANES 2010–2011	1000/770, ≥65	Protein intake (g/d) 24 h recall	Sarcopenia: ASM (by DXA)/Wt (%) < 2 SD below the mean of young health group without any chronic disease (20–39 years)	Lower protein intake in sarcopenia group compared to normal group in women and men (*p* = 0.13, 0.08)	NS	Inclusion

NS, no significant association; +, positive association. NO, number; FFQ, food frequency questionnaire; ASM, appendicular skeletal muscle mass; DXA, dual-energy X-ray absorptiometry; Ht, height; HGS, hand grip strength; AWGS, Asian Working Group for Sarcopenia; KNHANES, Korean National Health and Nutrition Examination Surveys; Wt, weight; SD, standard deviation; OR, odds ratio; CI, confidence interval; ref., reference; KFACS, Korean Frailty and Aging Cohort Study.

**Table 2 nutrients-16-04350-t002:** Characteristics of studies included in systematic review and meta-analysis on protein intake with sarcopenia-related indicators.

Author (Year)	Study Design (F/U)	Data Source	NO. of Subjects (Women/Men) Age (Years)	Exposure Variable Dietary Assessment Method	Outcome and Definition	Results	Main Finding	Meta-Analysis
Sarcopenic obesity								
Lee JH et al. (2021) [49]	Cross-sectional	KNHANES 2008–2011	2193/1635, ≥65	Protein intake (g/kg/d) 24 h recall	Sarcopenic obesity: ASM (by DXA)/BMI < 0.51 in women, <0.78 in men, and BMI ≥ 25 kg/m^2^	Lower protein intake in sarcopenic obesity group compared to non-sarcopenic obesity group in women and men (*p* < 0.001, *p* = 0.001)	Sarcopenic obesity +	Exclusion: Unable to synthesize (single outcome)
Yoo SJ et al. (2020) [50]	Cross-sectional	KNHANES 2008–2011	854/401 ≥65	Insufficient protein intake (<0.91 g/kg/d/) 24 h recall	Sarcopenic obesity: ASM (by DXA)/Wt (%) < 1 SD below the mean of young health group without any chronic disease (20–39 years) and BMI ≥ 25 kg/m^2^	Insufficient protein intake group’s OR 1.71, 95% CI 1.04–2.79 (ref. non-sarcopenic obesity)	Sarcopenic obesity +	Exclusion: No data on protein intake groups
Lim HS et al. (2018) [42]	Cross-sectional	KNHANES 2008–2011	1642/1850, ≥65	Protein intake (g/d) 24 h recall	Sarcopenic obesity: ASM (by DXA)/Wt (%) < 1 SD below the mean of young health group without any chronic disease (20–39 years) and WC > 85 cm in women, >90 cm in men	Higher protein intake in sarcopenic obesity group compared to sarcopenia group (*p* = 0.05)	Sarcopenic obesity +	Exclusion: No data on protein intake groups
Muscle mass								
Li X et al. (2022) [24]	RCT (8 wks)	Local community	11/13, ≥65	Protein supplement (20 g/d whey protein + 15 g/d soybean protein; includes 3 g branched-chain amino acid)	Skeletal muscle mass (kg): Measured by BIA (Inbody 370)	Skeletal muscle mass increase in the intervention group compared to the baseline (*p* < 0.01)	Skeletal muscle mass +	Inclusion
Na WR et al. (2021) [45]	RCT (90 days)	Facilities of community care	44/9, Mean ± SD 80.5 ± 7.0	Oral nutritional supplement (200 mL/d, 200 kcal, 15 g protein, with some micronutrients) Dietary records	Lean body mass (kg): Measured by BIA (Inbody S10) ASM (kg): Measured by BIA (Inbody S10)	Lean body mass improved in the intervention group (*p* = 0.01) ASM tends to increase in the intervention group (*p* = 0.18)	Lean body mass +	Exclusion: Unable to synthesize (different definition)
Park YS et al. (2018) [16]	RCT (12 wks)	Welfare centers	78/42, 70–85	Protein supplement (200 kcal/d, 0.5 g fat, 0.2 g cocoa powder, 9.3 g whey protein/10 g pack) provided to meet 1.2 or 1.5 g/kg/d 3-day dietary intake	ASM (kg): Measured by DXA	ASM is higher in 1.5 g/kg/d protein intake group compared to 0.8 g/kg/d group (*p* = 0.04)	ASM +	Exclusion: Unable to synthesize (different definition)
Kim YS et al. (2016) [47]	RCT (12 wks)	Local community	23/0, 65–74	Protein supplement (47 g/d Whey protein isolate, with some micronutrients) and resistance exercise	Skeletal muscle mass (kg): NA	Skeletal muscle mass improved in the training with protein group (*t* = 4.02)	Skeletal muscle mass +	Inclusion
Park JH (2006) [48]	Non-RCT (10 wks)	Welfare center facilities	27/0, Mean ± SD 73 ± 3	Protein food (200 mL of milk + 1 boiled egg 50 g) and resistance training Recommended daily diet	Muscle mass (kg): Measured by BIA (Inbody 4.0)	Muscle mass improved in the training with protein group (*p* < 0.05)	Muscle mass +	Exclusion: Unable to synthesize (different definition)
Han KH et al. (1997) [46]	Non-RCT (4 wks)	Care facility	25/0, ≥65	Oral nutritional supplement (New care 500 mL/d) 24 h recall	Lean body mass (kg): Wt (kg)—fat Wt (kg), measured by BIA (GIF-891)	Lean body mass tended to improve in the intervention group (*p* value NA)	NS	Exclusion: No comparison group
Oh C et al. (2018) [51]	Cross-sectional	KNHANES 2009	916/651, ≥65	Protein intake (g/kg/d) 24 h recall	Muscle mass (g): Measured by DXA ASM/Wt (%): Measured by DXA	Muscle mass, ASM/Wt (%) improved in protein intake 0.8–1.2 g/kg/d compared to <0.8 g/kg/d group (*p* < 0.001, *p* < 0.009)	Muscle mass + AMS/Wt (%) +	Exclusion: Unable to synthesize (different definition)
HGS								
Li X et al. (2022) [24]	RCT (8 wks)	Local community	11/13, ≥65	Protein supplement (20 g/d whey protein + 15 g/d soybean protein; includes 3 g branched-chain amino acid)	Relative grip strength in right and left: HGS (kg) (by a digital dynamometer)/Wt	Relative grip strength in right and left improved in the intervention group (*p* = 0.007, *p* = 0.001)	Relative grip strength +	Exclusion: No comparison group
Na WR et al. (2021) [45]	RCT (90 days)	Facilities of community care	44/9, Mean ± SD 80.5 ± 7.0	Oral nutritional supplement (200 mL/d, 200 kcal, 15 g protein, with some micronutrients) Dietary records	HGS (kg): Using a digital dynamometer	HGS tended to improve in the intervention group (*p* = 0.79)	NS	Exclusion: Unable to synthesize (different data types)
Park YS et al. (2018) [16]	RCT (12 wks)	Welfare centers	78/42, 70–85	Protein supplement (200 kcal/d, 0.5 g fat, 0.2 g cocoa powder, 9.3 g whey protein/10 g pack) provided to meet 1.2 or 1.5 g/kg/d 3-day dietary records	HGS (kg): Using a dynamometer	HGS improved over the intervention period in all group (*p* = <0.001)	HGS +	Exclusion: Unable to synthesize (different data types)
Kim CO et al. (2013) [23]	RCT (12 wks)	NHHS	69/15, ≥65	Oral nutritional supplement (Greenbia HP 400 mL/d, 400 kcal, 25 g protein; includes 9.4 g essential amino acids and some micronutrients) 24 h recall	HGS (kg): Using a dynamometer	No differences between groups	NS	Exclusion: Unable to synthesize (different data types)
Choi KA et al. (2022) [43]	Cross-sectional	Hip fracture patients	35/12, ≥65	Protein intake (g/d) FFQ	HGS (kg): Using a digital dynamometer	Lower protein intake associated with lower grip strength (*p* = 0.04)	HGS +	Exclusion: No comparison group
I JH et al. (2022) [52]	Cross-sectional	KNHANES 2014–2019	3401/2570, ≥65	Essential amino acid score (meeting the RNI for each essential amino acid) quartile 24 h recall	HGS (kg): Using a digital dynamometer High muscle strength: HGS ≥ 18 kg in women, ≥28 kg in men	HGS increase with essential amino acid score quartile (*p* = 0.001) Essential amino acid score Q4 group’s high muscle strength OR 1.38, 95% CI 1.07–1.79 (ref. Q1)	HGS +	Exclusion: Unable to synthesize (single outcome)
Park MY et al. (2022) [53]	Cross-sectional	KNHANES 2016–2018	0/1514, 65–80	Protein intake (g/kg/d) 24 h recall	Relative grip strength in dominant hand: HGS (kg) (by digital dynamometer)/Wt	Relative grip strength Q4 had higher protein intake (*p* < 0.0001)	Relative grip strength +	Exclusion: Unable to synthesize (single outcome)
Park SH et al. (2021) [54]	Cross-sectional	KNHANES 2014–2018	2728/2124, ≥65	Branched-chain amino acid intake (g/d) 24 h recall	HGS (kg): Using a digital dynamometer	Leucine intake Q4 group’s *β*-Coefficient ± SE 0.80 ± 0.38 (ref. Q1, *p* for trend = 0.03)	HGS +	Exclusion: No data on protein intake groups
Jang W et al. (2020) [55]	Cross-sectional	KNHANES 2016–2018	2083/0, ≥65	Quartiles of protein intake 24 h recall	Low HGS: Using a digital dynamometer, <18 kg in women	Protein group Q4’s OR 0.65, 95% CI 0.38–1.10 (ref. Q1)	NS	Inclusion
Kim CR et al. (2019) [56]	Cross-sectional	KNHANES 2014–2016	1942/1692, ≥65	Inadequate protein intake (40 g/d in women, 45 g/d in men) 24 h recall	Low HGS: Using a digital dynamometer, <28.9 kg in men, <16.8 kg in women	Inadequate protein intake group’s OR in men 1.50, 95% CI 1.05–2.15	Low HGS +	Inclusion
SPPB, Balance								
Li X et al. (2022) [24]	RCT (8 wks)	Local community	11/13, ≥65	Protein supplement (20 g/d whey protein + 15 g/d soybean protein; includes 3 g branched-chain amino acid)	SPPB score: sum of scores (gait speed, balance test, chair stand time) Balance score: Scoring side-by-side, semi-tandem, and tandem stands	No differences between group in SPPB, balance score	NS	Inclusion
Park YS et al. (2018) [16]	RCT (12 wks)	Welfare centers	78/42, 70–85	Protein supplement (200 kcal/d, 0.5 g fat, 0.2 g cocoa powder, 9.3 g whey protein/10 g pack) provided to meet 1.2 or 1.5 g/kg/d 3-day dietary records	SPPB score: sum of scores (gait speed, balance test, chair stand time) Balance (s): time for side-by-side, semi-tandem, and tandem stands	SPPB, balance improved over the intervention period in all group (*p* = <0.001, *p* = 0.002)	SPPB + Balance +	Inclusion
Kim CO et al. (2013) [23]	RCT (12 wks)	NHHS	69/15, ≥65	Oral nutritional supplement (Greenbia HP 400 mL/d, 400 kcal, 25 g protein; includes 9.4 g essential amino acids and some micronutrients) 24 h recall	SPPB score: sum of scores (gait speed, balance test, chair stand time) One-legged stance (s): time standing on one leg	SPPB score remained stable in the intervention group vs. decreased by 12.5% in controls (*p* = 0.04) No differences between group in one-legged stance	SPPB +	Inclusion
Gait speed, TUG								
Li X et al. (2022) [24]	RCT (8 wks)	Local community	11/13, ≥65	Protein supplement (20 g/d whey protein + 15 g/d soybean protein; includes 3 g branched-chain amino acid)	Gait speed score: scoring 4 m gait speed	No differences between group in gait speed score	NS	Exclusion: Unable to synthesize (different definition)
Park YS et al. (2018) [16]	RCT (12 wks)	Welfare centers	78/42, 70–85	Protein supplement (200 kcal/d, 0.5 g fat, 0.2 g cocoa powder, 9.3 g whey protein/10 g pack) provided to meet 1.2 or 1.5 g/kg/d 3-day dietary intake	Gait speed (m/s): time for 4 m walk TUG (s): time to stand up from a chair, walk 3 m, turn around, walk back, and sit down again	Gait speed improved in 1.5 g/kg/d protein intake group compared to 0.8 g/kg/d group (*p* = 0.04) TUG improved over the intervention period in all group (*p* = <0.001	Gait speed + TUG test +	Inclusion
Kim CO et al. (2013) [23]	RCT (12 wks)	NHHS	69/15, ≥65	Oral nutritional supplement (Greenbia HP 400 mL/d, 400 kcal, 25 g protein; includes 9.4 g essential amino acids and some micronutrients) 24 h recall	Gait speed (m/s): time for 4 m walk TUG (s): time to stand up from a chair, walk 3 m, turn around, walk back, and sit down again	Gait speed decreased by 1.0% in the intervention group vs. 11.3% in controls (*p* = 0.04) TUG improved by 7.2% in the intervention group vs. declined by 3.4% in controls (*p* = 0.04)	Gait speed + TUG test +	Inclusion

NS, no significant association; +, positive association; F/U, follow-up; NO, number; KNHANES, Korean National Health and Nutrition Examination Surveys; ASM, appendicular skeletal muscle mass; DXA, dual-energy X-ray absorptiometry; BMI, body mass index; Wt, weight; SD, standard deviation; OR, odds ratio; CI, confidence interval; ref., reference; WC, waist circumference; RCT, randomized controlled trial; BIA, Bioelectrical Impedance Analysis; NA, not available; HGS, hand grip strength; NHHS, National Home Healthcare Services; FFQ, food frequency questionnaire; RNI, recommended nutrient intake; SE, standard error; SPPB, short physical performance battery; TUG, timed up-and-go; vs., versus.

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
