# Peer review of "Association of Protein Intake with Sarcopenia and Related Indicators Among Korean Older Adults: A Systematic Review and Meta-Analysis"

_nutrients, 2024, doi:10.3390/nu16244350_

Round 1

Reviewer 1 Report

Comments and Suggestions for Authors

Han et al investigated the protein intake and prevalence of sarcopenia in older Korean population. The rationale is reasonable, materials and methods are well defined, results and discussion are well illustrated/relevant to the topic. The most important take home message is that - the risk of sarcopenia was significantly higher in the < 0.8 g/kg/day protein intake group rekative to the group with 0.8-1.2 g/kg/day or ≥1.2 g/kg/day protein intake group. This is not too surprised to the general audiences. Their results are not different from the experts in this field and thus the clinical application of this research is limited. Quite a few recent articles have shown the similar results (e.g., Coelho-Junior et al., Protein Intake and Sarcopenia in Older Adults: A Systematic Review and Meta-Analysis. Int. J. Environ. Res. Public Health 2022, 19, 8718. https://doi.org/10.3390/ ijerph19148718. Tu et al., Sarcopenia among the Elderly Population: A Systematic Review and Meta-Analysis of Randomized Controlled Trials. Healthcare 2021, 9, 650. https://doi.org/10.3390/healthcare9060650.)

Further comments

On page 5, line 194, 9 articles were included in the meta-analysis, however, on table 1, only 8 articles were listed.

For the analysis, is it possible to separate Koren women from men? And what will be the results? I look at the actual number of subjects listed in table 1, n<10,000 for women and men, respectively. With such a small number, the clinical relevance is very restricted, 

Reviewer 2 Report

Comments and Suggestions for Authors

  nutrients-3345734 Association of protein intake with sarcopenia and related indicators among Korean older adults: a systematic review and meta-analysis Minjee Han , Kyung-sook Woo , Kirang Kim  Overall: This Systematic Review confirms previous meta-analyses on the role of adequate protein intake to mitigate sarcopenia in older adults. While the study is not novel per se and the number of studies included are very limited (major drawback), the PRISMA model has been applied, and the results are largely in support of definite meta-analyses performed in other populations.Minor:  Please provide a more in-depth discussion regarding why some of the individual components (such as low LBM/muscle mass, SPPB score, walk speed, and TUG) were not significantly associated with protein intake. Why do you propose handgrip was significant, while lower body function/strength (SPPB, walk speed, TUG) was not? Are you arguing that some diagnostic criteria should be performed over others in Korean populations?  Best of luck in your future research, 

Reviewer 3 Report

Comments and Suggestions for Authors

General Comments

The manuscript addresses an important topic and provides valuable insights into the relationship between protein intake and sarcopenia in older adults in Korea. It is well-structured, follows PRISMA guidelines, and presents robust statistical analyses. However, certain areas need refinement for clarity, consistency, and broader applicability.

Specific Comments

Title and Abstract

Line 1–26 (Abstract):

The objectives are clear, but the conclusion could emphasize actionable recommendations, such as specific protein intake guidelines for Korean older adults.

Suggested revision: Include a brief mention of the need for combined interventions (e.g., protein intake and exercise) to strengthen sarcopenia prevention efforts.

Introduction

Line 28–80:

Line 35–40: The prevalence of sarcopenia among Korean older adults is well-described but lacks global context. Adding a comparison with international prevalence rates can enhance the relevance of the study.

Line 41–55: The section discusses optimal protein intake standards but does not clearly explain the physiological basis for the 1.2 g/kg/day recommendation. Expand on how this threshold aligns with the anabolic resistance seen in aging.

Methods

Line 81–183:

Line 86–91: The search strategy is thorough, but the keywords used (e.g., "physical functional performance") might miss relevant studies with different terminologies. Consider broadening the keyword list or explaining the rationale for the chosen terms.

Line 94–98: The inclusion of Korean databases is a strength; however, a brief mention of why certain databases (e.g., Web of Science) were excluded would add transparency.

Line 143–149: The process of standardizing protein intake (e.g., grams per body weight) is well-explained, but the assumptions made (e.g., using average body weight from KNHANES data) should be explicitly stated.

Line 180–183: Clarify why ethical approval was not deemed necessary, especially if secondary data analysis involved sensitive population groups.

Results

Line 184–239:

Line 186–194 (Study selection): The inclusion of 23 studies is commendable, but the flowchart (Figure 1) could benefit from more detailed labels for excluded studies (e.g., "insufficient data on protein intake").

Line 200–213 (Characteristics): The diagnostic criteria for sarcopenia vary across studies, which is acknowledged. However, a table summarizing the criteria used in each study would provide clarity and facilitate comparison.

Line 216–236 (Meta-analysis results):

The use of fixed-effects models is justified by small between-study variance, but with I² values of up to 71% (Line 55), random-effects models could have provided additional robustness.

Figure 2: Add clearer legends explaining the clinical relevance of OR values for each protein intake group.

Discussion

Line 146–238:

Line 155–161: The comparison with international findings (e.g., Health ABC Study) is valuable. However, more emphasis on unique dietary patterns of Korean older adults (e.g., higher reliance on plant protein) would strengthen the cultural context.

Line 164–174: The lack of significant findings for some indicators (e.g., skeletal muscle mass, SPPB) should be discussed in relation to the short intervention durations (4–12 weeks) in the included RCTs.

Line 180–189: The discussion on current guidelines (e.g., 0.8–1.2 g/kg/day) could be expanded to highlight gaps in evidence specific to older adults with chronic conditions.

Limitations

Line 217–234:

The limitations are well-articulated, but the following should be expanded:

Line 217–222: Explain how variations in diagnostic criteria might influence the comparability of results.

Line 223–227: Discuss potential biases introduced during unit standardization of protein intake.

Line 228–234: Consider including a note on the challenges of using cross-sectional data for causal inferences.

Figures and Tables

Figures 2 and 4: Improve legends by explaining how the OR values translate into practical dietary recommendations.

Table 1 (Study Characteristics): Add a column summarizing the main findings of each study for quick reference.

References

Ensure consistency in formatting (e.g., DOI links).

Expand references to include more global studies on protein intake and sarcopenia, providing broader context.

Summary of Recommendations

Expand the discussion of cultural dietary patterns and their implications.

Provide a table summarizing sarcopenia diagnostic criteria used in the included studies.

Justify methodological choices (e.g., fixed-effects models) given the observed heterogeneity.

Highlight actionable dietary recommendations in the conclusion.

Add clarity to figures and legends for better interpretation of results.

The manuscript makes a valuable contribution to the field but would benefit from addressing the above suggestions to improve clarity, methodological rigor, and practical relevance.

Round 2

Reviewer 1 Report

Comments and Suggestions for Authors

No further comments. The authors attend to address my concerns raised in the previous version of manuscript. 

Reviewer 3 Report

Comments and Suggestions for Authors

All concerns have been addressed.